# Accelerating target deconvolution for therapeutic antibody candidates using highly parallelized genome editing

Jenny Mattsson[1,2], Ludvig Ekdahl[1], Fredrik Junghus[1], Ram Ajore[1], Eva Erlandsson[1], Abhishek Niroula [1,3], Maroulio Pertesi [1], Björn Frendéus[2], Ingrid Teige[2] & Björn Nilsson [1,3]✉

Therapeutic antibodies are transforming the treatment of cancer and autoimmune diseases. Today, a key challenge is finding antibodies against new targets. Phenotypic discovery promises to achieve this by enabling discovery of antibodies with therapeutic potential without specifying the molecular target a priori. Yet, deconvoluting the targets of phenotypically discovered antibodies remains a bottleneck; efficient deconvolution methods are needed for phenotypic discovery to reach its full potential. Here, we report a comprehensive investigation of a target deconvolution approach based on pooled CRISPR/Cas9. Applying this approach within three real-world phenotypic discovery programs, we rapidly deconvolute the targets of 38 of 39 test antibodies (97%), a success rate far higher than with existing approaches. Moreover, the approach scales well, requires much less work, and robustly identifies antibodies against the major histocompatibility complex. Our data establish CRISPR/Cas9 as a highly efficient target deconvolution approach, with immediate implications for the development of antibody-based drugs.

---

[1] Department of Laboratory Medicine, Hematology and Transfusion Medicine, Lund, Sweden. [2] BioInvent International AB, Ideongatan 1, Lund, Sweden. [3] Broad Institute, 415 Main Street, Cambridge, MA, USA. ✉email: bjorn.nilsson@med.lu.se

Monoclonal antibodies (mAbs) are a successful class of drugs that have drastically improved the treatment of cancer and autoimmune diseases[1,2]. Yet, many mAbs currently used in the clinic target the same antigen, and the commercial competition focuses on less than a dozen targets, including TNF-α, CD20, HER2, EGFR, VEGFA, PD-1/PD-L1, and IL6/IL6R[2,3].

For these reasons, there is today a strong interest in developing therapeutic mAbs against entirely new targets. This has created a broad interest in phenotypic discovery (PD) strategies[4–6]. Unlike traditional target-based discovery, PD enables searches for mAbs with therapeutic potential without specifying a molecular target a priori. In most therapeutic contexts, PD is employed to identify mAbs that target a relevant cell type (e.g., cancer cells or immune cell subsets). Technically, PD can be achieved by incubating cells with antibody phage-display libraries, selecting antibodies that bind target cells, and subsequently identifying mAbs with desired function (e.g., cell killing or immune cell activation)[7,8].

While PD enables discovery of mAbs against unexpected targets in principle, many researchers shy away from such strategies because of the challenges associated with target deconvolution[9]. Since the mAb target is unknown a priori, it needs to be identified (deconvoluted) downstream. The problem is that existing deconvolution methods, mainly immunoprecipitation[10] and protein library overexpression[11], are notoriously time- and resource-consuming, unreliable, and scale poorly with the number of antibodies[4–6,10–15]. Efficient deconvolution methods are eagerly needed to allow a broader use of PD, and thereby faster development of antibody-based drugs.

In this work, we investigate if highly parallelized CRISPR/Cas9 screening can be employed to deconvolute antibody targets efficiently. Hence, we explore an approach where antigen-positive test cells (i.e., cells that bind the antibody-of-interest) are transduced with a lentiviral sgRNA/Cas9 library to produce a cell pool harboring knockouts of known protein-coding genes (Fig. 1a). If the library includes sgRNAs towards the gene encoding the antibody's target protein, and this gene is not essential for the survival of the test cells[16], the transduced cell pool can be expected to contain a small subpopulation that has turned antigen-negative (if complete knockout) or antigen-weak (if incomplete knockout). Additionally, loss of antibody binding can be expected for cells that harbor knockouts of genes needed for the expression of the target protein, such as chaperones, partner proteins in multimeric complexes, or key transcription factors. Thus, both the antibody target and its dependencies can potentially be found by staining transduced cells with antibody-of-interest, enriching antigen-negative cells by cell sorting, and identifying sgRNA sequences enriched in these cells using massively parallel sequencing of integrated pro-lentiviral DNA (Fig. 1a). Applying this approach within three real-world phenotypic discovery programs, we deconvolute the targets of 38 of 39 test antibodies (97%). Moreover, we find that the approach scales well, requires much less work, and robustly identifies antibodies against the major histocompatibility complex. In conclusion, our results establish CRISPR/Cas9 as a highly efficient antibody target deconvolution approach.

## Results

### Test antibodies for target deconvolution

We decided to carry out a comprehensive investigation of the applicability, efficiency, and robustness of CRISPR/Cas9 as a target deconvolution approach. Accordingly, we applied our approach to 37 mAbs with unknown specificities and two with known specificities (CD2 and CD45). The former were developed by our team in three PD programs aimed at discovering new mAbs for cancer therapy, and

were isolated from the n-CoDeR antibody phage display library[17] through affinity selection against target cells and depletion with non-target cells followed by conversion from single-chain variable fragment to human IgG1 format[7]. In our test set, we first included 23 mAbs from a program aimed at finding mAbs against targets on regulatory T cells (Tregs) through phenotypic screening on primary Tregs isolated from cancer patients' ascites. Tregs attenuate anti-tumor immune responses and are currently targeted by mAbs against PD-1/PD-L1 and CTLA-4[18–21]. While Tregs cannot be cultured ex vivo in the numbers needed for pooled CRISPR/Cas9, our antibodies (denoted mAb1 to mAb23) bound the Jurkat or H9 T-cell lines, or primary CD4$^+$ T cells from healthy blood donors activated in vitro to induce a Treg-like phenotype (Supplementary Fig. 1). Second, we included six mAbs against tumor-associated macrophages (TAMs) through phenotypic screening on primary TAMs isolated from ascites. TAMs also regulate anti-tumor immune responses[22]. These antibodies (mAb24 to mAb29) bound the monocyte cell line THP-1, either with or without polarization towards a macrophage-like phenotype using phorbol 12-myristate 13-acetate (Supplementary Fig. 1). Third, we included eight antibodies developed against the prostate cancer cell line DU145 (mAb30 to mAb37) that were coincidentally found to bind Jurkat or H9 cells (Supplementary Fig. 1). Thus, we compiled a set of test mAbs that represent three real-world, cancer-related PD programs, target several different cell types, and are likely to exhibit a broad range of specificities.

### Deconvolution of targets

We transduced test cells with a genome-wide CRISPR/Cas9 knockout library, either GeCKO[23] (six sgRNA per gene; 125,411 in total) or Brunello[24] (four sgRNA per gene; 77,441 in total) (Supplementary Table 1). We stained transduced, puromycin-selected cells with test mAb and fluorescence-labeled detection antibody (Supplementary Fig. 1) and sorted the 1.5 to 2% least fluorescent cells to enrich antigen-negative cells. To maintain library coverage, we used at least 100 cells per sgRNA in each sort (i.e., at least 8 million Brunello-transduced or 13 million GeCKO-transduced cells). As control cells, we used the 20% most fluorescent cells (17 mAbs) or unsorted cells (22 mAbs; Supplementary Table 1). For mAbs binding Jurkat, H9, and non-polarized THP-1 cells (which can be cultured after sorting), we performed two consecutive sorts separated by ex vivo expansion of sorted cells, in order to pre-enrich antigen-negative cells and increase the signal-to-noise ratio (Fig. 1b–d). From sorted cells, we extracted DNA and sequenced the sgRNA-encoding region. Using a robust ranking aggregation algorithm (MAGeCK; ref. [25]), we calculated an enrichment score for each gene that quantifies the representation of its targeting sgRNA sequences in antigen-negative cells relative to control cells. For each mAb, we analyzed 3 to 5 replicates. These were sort replicates from the same transduced cell pool for cell lines, and transduction replicates with cells from different donors for primary cells. For most antibodies, we observed high positive correlation in gene enrichment scores between replicates (Fig. 1e).

To prioritize target genes, we used three criteria: Firstly, we prioritized genes encoding membrane proteins with significant MAGeCK scores (FDR < 5%), identifying a single gene for each of 24 mAbs and two genes for each of 9 mAbs (Fig. 2a, Supplementary Fig. 2 and Supplementary Table 1). Interestingly, the latter pairs of genes were always functionally linked, for example integrin alpha and beta chains (Supplementary Table 1).

Secondly, we developed a gene set enrichment testing technique to recognize mAbs that bind the Major Histocompatibility Complex (MHC). In PD, antibodies are often selected by positive and negative selection with cells from different donors, each with his/her own sets of HLA alleles. Hence, MHC

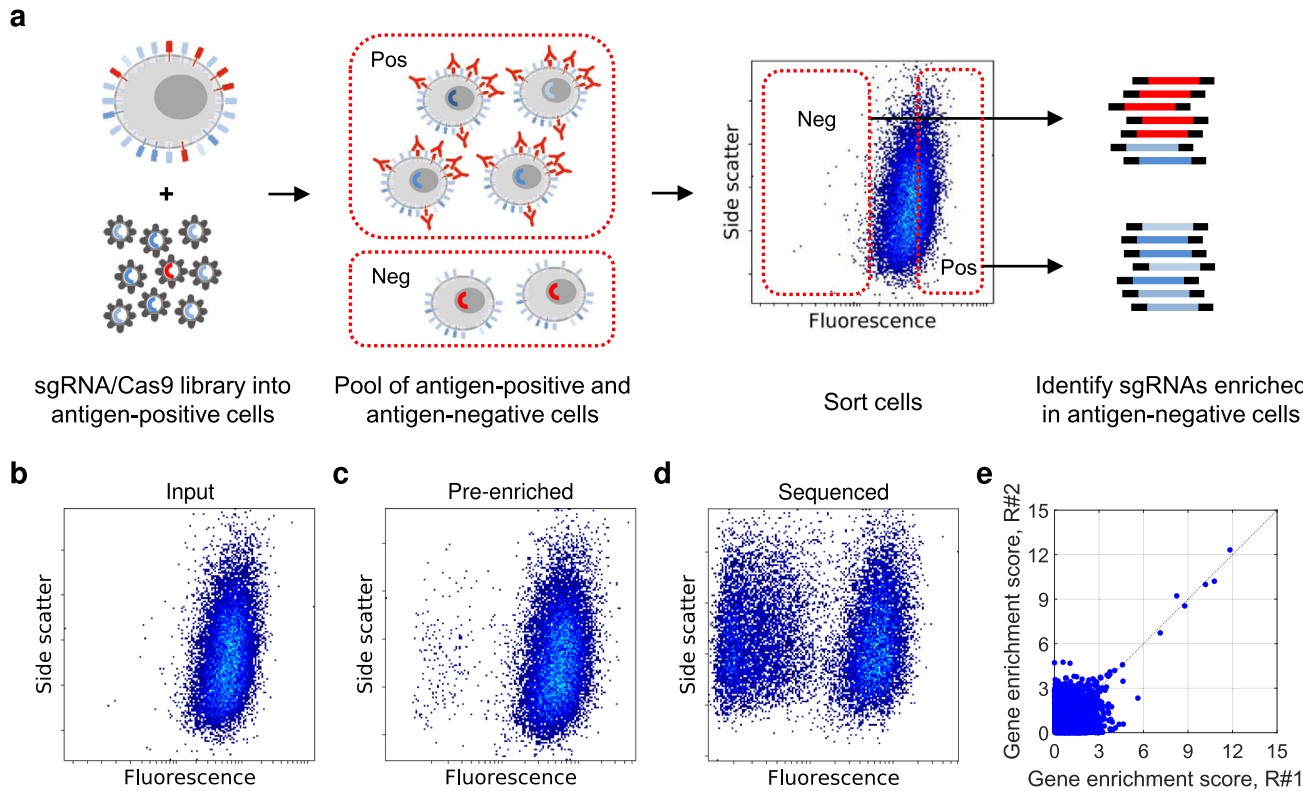

**Fig. 1 Antibody target deconvolution using CRISPR/Cas9 screening. a** Schematic outline of the target deconvolution process. Cells staining positive with the antibody of interest are transduced with a lentiviral sgRNA/Cas9 knockout library resulting in a heterogenous cell pool with a small population of antigen-negative cells. These cells with gene knockouts leading to lost or diminished antibody binding are isolated by FACS, the genomic DNA is extracted, and the sgRNA-encoding DNA is sequenced on the Illumina NextSeq 500 platform. Genes with sgRNAs enriched in the antigen-negative cells are identified, resulting in a proposed antibody target. **b–d** Representative example of flow cytometry data for mAb binding to transduced cells detected with anti-human IgG-APC. **b** Input cells before sort. **c** Pre-enriched cells after one sort with a distinct fraction of antigen-negative cells. **d** Sequenced cell pool after two sorts with a highly enriched fraction of antigen-negative cells. **e** Representative example of MAGeCK gene enrichment scores for two sort replicates, showing that the same genes are identified for both replicates. Sorts were performed in 3 to 5 replicates. Source data are provided as a Source Data file.

antibodies are commonly found in PD programs, particularly against MHC class I. To recognize such antibodies, we defined a set of 11 genes that are required for MHC class I expression[26–29], including MHC class I itself (*HLA-A*, *HLA-B*, *HLA-C* and *B2M*), key components of the peptide loading complex (*TAP1*, *TAP2* and *TAPBP*) and key transcription factors (*NLRC5*, *RFXAP*, *RFX5* and *RFXANK*). Hypothesizing that overrepresentation of high MAGeCK scores within this set could indicate MHC class I specificity, we defined a MHC class I enrichment score as the $-\log_{10}$ P-value for one-sided Wilcoxon rank-sum test for the MAGeCK scores for our 11 genes vs all other genes. Notably, we hereby address the problem that a mAb that binds two or more of the HLA-A, B and C alpha chains may not be detected via sgRNAs targeting, say, HLA-A, as residual binding to HLA-B or HLA-C can leave the cells antigen-positive. Thus, sgRNAs against a gene required for expression of all three alpha chains, could more reliably reveal that the mAb targets the MHC. In total, 13 test mAbs showed a significant MHC class I enrichment score (Bonferroni-adjusted Wilcoxon $P < 0.05$; Fig. 2a-b, Supplementary Fig. 2 and Supplementary Table 1).

Lastly, for the four remaining mAbs where we could not prioritize a target gene using our first two criteria (i.e., no significant membrane protein gene or enrichment of MHC dependencies), we prioritized the highest-ranked membrane protein gene even if not significant (Fig. 2a, Supplementary Fig. 2 and Supplementary Table 1).

**Validation of inferred specificities.** To validate the inferred specificities, we carried out extensive ELISA, blocking and overexpression experiments, confirming the specificity of 23 of the 24 identified non-MHC antibodies and 9 of the 13 MHC class I antibodies (Supplementary Fig. 3 and Supplementary Table 2). The lower validation rate for MHC antibodies is expected as the high genetic variability for MHC class I (17,000 alleles identified to date[30]) complicates the confirmation of MHC I specificity. Importantly, however, we noted a significant difference in MHC class I enrichment scores between the 23 validated non-MHC antibodies and the 9 validated MHC antibodies (Fig. 2c), confirming our hypothesis that enrichment of MAGeCK signal among our 11 MHC class I dependency genes indicates MHC specificity. Moreover, the four mAbs with inferred MHC class I specificity that could not be validated experimentally showed MHC class I enrichment scores similar to the nine MHC class I antibodies that did validate experimentally. Hence, it is reasonable to conclude that these four mAbs also target MHC class I. In summary, our CRISPR/Cas9 approach identified the targets of 36 of the 37 mAbs with unknown specificities, plus the targets of the two control antibodies (Supplementary Table 2). This yields a success rate of 38/39 (97%), leaving only one mAb unresolved (mAb32). Strikingly, for 34 of the resolved mAbs, a gene representing the target was the highest-ranking gene in their respective CRISPR/Cas9 data (Fig. 2d).

Among the deconvoluted targets, we noted several that are applicable to immunotherapy. For example, CD25 (targeted by

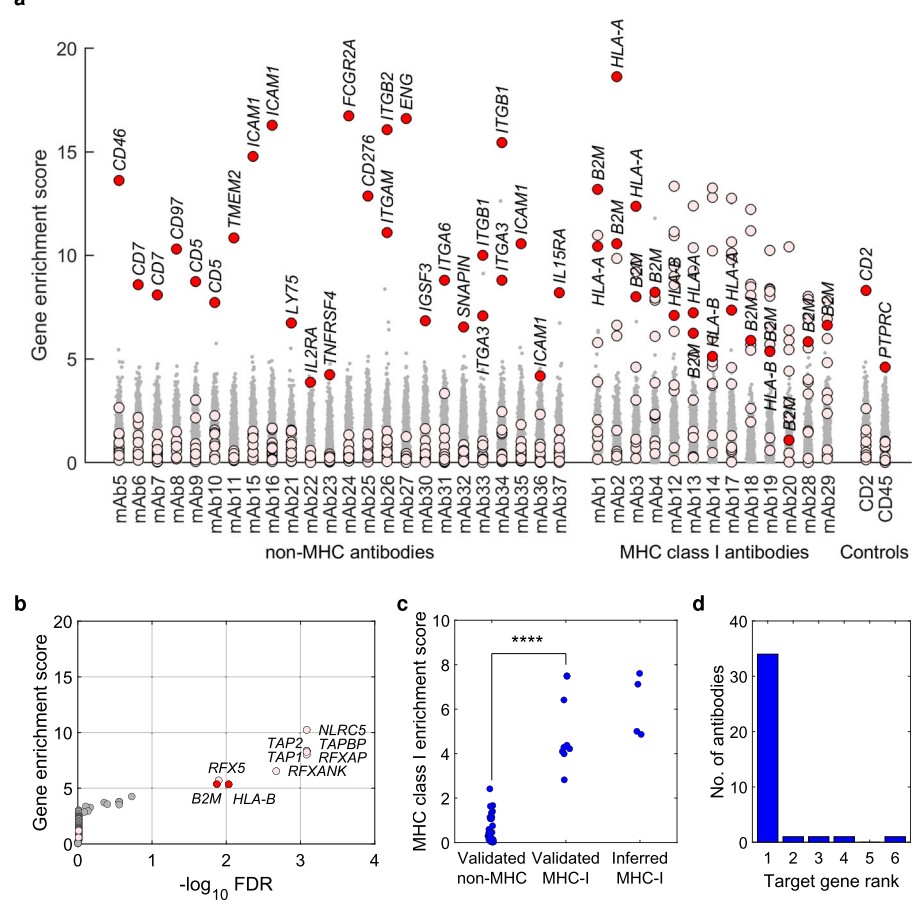

**Fig. 2 Identification of antibody targets. a** MAGeCK gene enrichment scores for all genes across the 39 test mAbs. Target genes selected using our three criteria indicated in red, MHC class I dependency genes in pink. **b** Representative example showing enrichment of high MAGeCK scores among MHC class I dependency genes for a mAb against the MHC class I complex (mAb19). MAGeCK false discovery rate (x-axis) vs gene enrichment score for identified MHC class I specific antibody mAb19 (y-axis). Target genes selected using our three criteria indicated in red, MHC class I dependency genes in pink. **c** MHC class I enrichment score, calculated as $-\log_{10}(P\text{-value})$ for one-sided Wilcoxon rank-sum test for MAGeCK scores for our 11 selected MHC class I dependency genes vs other genes in the genome. As shown, we observed significantly higher enrichment scores for mAbs confirmed to target MHC class I ($n = 9$) compared to antibodies with non-MHC targets ($n = 25$; **** Indicates $P < 0.0001$ by one-sided Wilcoxon rank-sum test; exact $P$-value $5.5 \times 10^{-6}$). Based on this observation, we infer that four antibodies (right) that showed high MHC class I enrichment scores but whose specificity could not be validated experimentally ($n = 4$) also target MHC class I. **d** Histogram of the rankings of the best-scoring gene representing the target in the MAGeCK analysis across the 38 resolved mAbs. Source data are provided as a Source Data file.

mAb22) has been proposed as a target molecule for Treg depletion[31,32], and ox40 (mAb23) a target for augmenting anti-tumor immune responses by activating effector T-cells[33–35]. Additionally, targeting of B7-H3, Mac-1 and Endoglin (mAb25, mAb26, and mAb27) has been reported to polarize TAMs from their common, pro-tumorigenic ("M2") state towards an anti-tumorigenic ("M1") state, potentially enhancing anti-tumor immune responses[36–38]. B7-H3 is an immune-checkpoint molecule that plays an important role in the inhibition of TAM and T cell function, is highly over-expressed on a wide range of human solid cancers and often correlate with unfavorable outcome, and antibodies against B7-H3 are currently in clinical trials[39]. Activation of the Mac-1 α-chain CD11b suppress tumor-growth in pre-clinical studies[37,40]. Finally, low expression of Endoglin is associated with improved clinical outcome in several cancers[41,42], and cancer patients with concurrent hereditary hemorrhagic telangiectasia (HHT), a hereditary condition caused by deleterious *ENG* mutations show longer survival[43]. Endoglin antibodies are currently under development[41,44]. These examples illustrate the ability of

CRISPR/Cas9 to deconvolute the targets of PD antibodies with therapeutic potential.

**Antibody target dependencies**. Unlike existing deconvolution approaches, CRISPR/Cas9 screening can uncover target-related biology via dependencies, not only for MHC class I but also for other targets. We therefore searched our data for genes with a significant MAGeCK score (FDR < 5%) and known biological connections to verified targets. For three integrin-targeting antibodies (mAb26, mAb33 and mAb34), we found not only the genes encoding their α- and β-chains, but also dependencies on *HSP90B1* and *MESDC2* (Supplementary Fig. 2). Interestingly, *HSP90B1* encodes gp96, required for the expression of most integrins[45] and *MESDC2* encodes a chaperone that interacts with gp96[46,47]. Additionally, for one integrin antibody (mAb26), we identified dependency on *MAP2K1* (Supplementary Fig. 2), which influences integrin surface expression[48]. Finally, for an antibody targeting IL15RA (mAb37), we identified a dependency on the transcriptional factor IRF2, which is known to modulate IL15RA

 

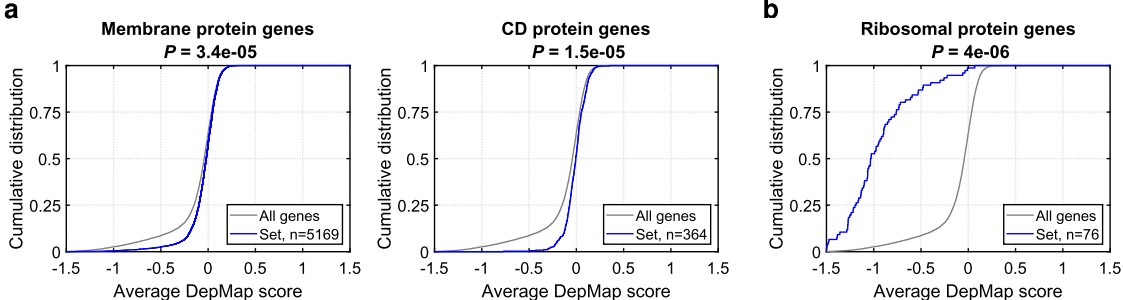

**Fig. 3 DepMap scores showing gene essentiality.** Since deconvolution by CRISPR/Cas9 requires that the target protein encoding gene is not essential for survival of the test cells, we investigated if cell membrane genes are more, or less, essential than genes in general. To this end, we used dependency scores from CRISPR/Cas9 proliferation screens for 436 cell lines from DepMap. In essence, a negative DepMap score indicates that a gene is essential for the survival/proliferation of a large fraction of cell lines. Gray lines indicate all genes, blue lines investigated gene set. **a** Distributions of DepMap scores for membrane protein genes and cluster designation (CD) antigen genes. As shown, these gene sets are significantly depleted of negative DepMap scores relative to other genes in the genome, indicating that they are less essential. This observation indicates that failure to deconvolute a target because of essentiality is unlikely. **b** By contrast, gene sets that are essential for cell survival are enriched for negative DepMap scores, exemplified here by ribosomal protein genes. In all panels, *P*-values for enrichment/depletion were calculated using RenderCat using default settings (two-sided Zhang C goodness-of-fit test).

expression[49] (Supplementary Fig. 2). These results illustrate the ability of library-based CRISPR/Cas9 to illuminate target biology through dependencies.

**Essentiality of surface protein genes for cell survival.** Finally, since deconvolution by CRISPR/Cas9 requires that the target protein encoding gene is not essential for survival of the test cells, we asked if cell surface proteins are more, or less, essential than genes in general. To this end, we analyzed data from genome-wide sgRNA proliferation screens encompassing 436 cell lines from the Dependency Map (DepMap)[16]. A DepMap gene score reflects the representation of sgRNAs targeting a gene after three weeks of culture, compared to the representation of the same sgRNAs in the original library. A negative score means that the gene is essential. We observed that sets of genes encoding membrane proteins and cluster designation (CD) antigens contain fewer negative DepMap scores compared to other genes in the genome[50] (Fig. 3a). This is in contrast to classes of genes that are known to be essential for cell survival and hence enriched for negative DepMap scores, for example ribosomal protein genes (Fig. 3b). Since PD antibodies are mainly developed against membrane proteins on cell surfaces, these data indicate that failure to deconvolute an antibody target because of gene essentiality is unlikely.

## Discussion

The adoption of PD into real-world antibody development has been slow due to the lack of efficient target deconvolution methods[9]. Here, we show that pooled CRISPR/Cas9, coupled to cell sorting and massively parallel single-molecule sequencing, provides a fast and robust way to deconvolute the targets of cell surface antibodies.

In contrast to immunoprecipitation which is time-consuming and unreliable[5,6,10,14,15], deconvolution using pooled CRISPR/Cas9 can be done in a few weeks in an experienced lab, with a high chance of success (here 97%). With immunoprecipitation, deconvoluting even a single antibody requires significant effort (order of months at best in the typical case), and the success rate is depressing. CRISPR/Cas9, on the other hand, scales well and can be readily applied to reasonably sized sets of antibodies without specific optimization. The scalability is important as PD often involve the deconvolution of several dozens of antibodies, as illustrated here. Moreover, unlike immunoprecipitation, our

approach does not seem to require high target abundance (strong staining) but seems to work for targets of intermediate abundance (intermediate staining; c.f., mAb31 and mAb37 in Supplementary Fig. 1). The lower limit for abundance (staining) remains to be systematically determined. However, we expect that the lower the abundance is, the lower the signal-to-noise ratio will become, though this can be partly overcome by increasing the number sorting/expansion rounds (Fig. 1b–d), and the number of replicates (i.e., the statistical power).

Further, our approach offers advantages over library over-expression techniques. Firstly, these techniques lack the ability to identify antibodies towards protein complexes, or obtain insight into the underlying pathways, through dependencies the way pooled CRISPR/Cas9 does. Secondly, arrayed overexpression reagents are hard to make, practically only available at a high cost via commercial services, and it is well known that the chance of success is considerably lower than in our study[5,6,11,14]. Thirdly, pooled cDNA library reagents[51,52] also pose challenges that are not present with Cas9/sgRNA libraries, including being more difficult to synthesize, transfect and deconvolute. Finally, while pooled CRISPR-dCas9/sgRNA overexpression libraries have been developed, these produce effect sizes that are significantly weaker than complete knockouts[53,54].

Finally, our data demonstrate that pooled CRISPR/Cas9 can identify targets not only directly, but also indirectly through dependencies. We illustrate the indirect approach by developing a gene set enrichment testing technique to aid the deconvolution of MHC class I antibodies (Fig. 2c), and predict that the use of dependencies can be extended to aid deconvolution of antibodies against other protein complexes. For example, our data suggest that dependencies could be used to aid the deconvolution of integrin antibodies, as the genes *HSP90B1* and *MESDC2* were called significant for three such antibodies (mAb26, mAb33, mAb34; Supplementary Fig. 2).

Regarding limitations, our approach requires a test cell that is antigen-positive and can be cultured and transduced. For most applications, finding a workable test cell type should be uncomplicated. We expect that, for most antibodies, an antigen-positive cell line will work, even if the phenotypic screen itself has been done on primary cells. For example, 29 of the 38 antibodies in our test set were identified through phenotypic screens on primary cells (the 23 Treg and the 6 TAM antibodies). In case an appropriate antigen-positive cell line cannot be found, our approach can be applied to primary cells (in our case in vitro-

 

activated CD4$^+$ T cells from healthy blood donors; mAb18 to mAb23). If no antigen-positive, practically useable test cell can be identified, our approach is not applicable. A second, theoretical caveat is that targets that are essential for test cell survival could escape detection. However, our analysis (Fig. 3) shows that genes that encode membrane proteins are less essential on average, making target drop-out unlikely. A third limitation is that broadly reactive antibodies (i.e., those that target multiple, structurally similar proteins) will be difficult to deconvolute as the target genes are knocked-out one at the time, and the test cells will therefore never turn completely antigen-negative. A fourth limitation is that the approach is based on gene knock-out, and hence cannot differentiate between slightly different antibodies that target different epitopes on the same target molecule.

Moving forward, one refinement will be to develop CRISPR/Cas9 knockout libraries focused on genes that encode cell surface proteins, and their key dependencies. The advantage would be that fewer cells will be needed to maintain sgRNA library representation. With smaller libraries, the deconvolution experiment can be downscaled, reducing the cell culture work, sorting time, sequencing costs, and facilitating deconvolution in primary cells.

In conclusion, target deconvolution has been the main bottleneck of PD for years. Now, our results establish pooled, lentiviral CRISPR/Cas9 as a fast and robust way to deconvolute the targets of cell surface antibodies, potentially allowing a broader use of PD and thereby accelerating the discovery of antibody-based cancer therapeutics.

## Methods

**Cell culture**. Cell lines 293 T/17 (ATCC #CRL-11268), CHO-S (Thermo Fisher Scientific #R80007), Jurkat (DSMZ #ACC 282), H9 (ATCC #HTB-176) and THP-1 (ECACC #88081201) were cultured according to suppliers' instructions. 293 T/17 cells in DMEM (Thermo Fisher Scientific) supplemented with 10% Gibco Fetal Bovine Serum (FBS; Thermo Fisher Scientific), CHO-S cells in FreeStyle CHO Expression Medium (Thermo Fisher Scientific) supplemented with 8 mM Glutamine (Thermo Fisher Scientific) and remaining cells in RPMI 1640 GlutaMAX (Thermo Fisher Scientific) supplemented with 10% FBS, 10 mM HEPES (Thermo Fisher Scientific) and 1 mM sodium pyruvate (Thermo Fisher Scientific) (R10).

THP-1 cells were polarized by addition of 50 ng/mL of Phorbol 12-myristate 13-acetate (PMA; Sigma-Aldrich) to $2.5 \times 10^5$ cells/ml in 50 ml R10 medium in T-175 flasks. Cells were incubated for 24 h, culture medium was replaced with 50 ml fresh non-PMA supplemented medium and cells were incubated for a further 24 h. After incubation, cells were harvested with Accutase (Thermo Fisher Scientific).

Buffy coats from healthy donors (Hallands hospital, Halmstad) were collected in accordance with ethical permission from the Ethics Committee of Skåne University Hospital (2010/356) under informed consent. Peripheral Blood Mononuclear Cells (PBMC) were prepared from buffy coats by gradient density centrifugation (Ficoll Paque PLUS, GE Healthcare). CD4$^+$ T cells were positively selected using CD4 Microbeads (Miltenyi Biotec) according to manufacturer's instructions. Purified cells were activated using Dynabeads Human T-Activator CD3/CD28 (Thermo Fisher Scientific) according to manufacturer's instructions and cultured in R10 supplemented with 50 ng/ml IL-2 (Miltenyi Biotec).

**Amplification of CRISPR libraries**. Human GeCKOv2 CRISPR knockout pooled library was a gift from Feng Zhang (Addgene #1000000048) and human Brunello CRISPR knockout pooled library was a gift from David Root and John Doench (Addgene #73179). Library plasmids (1 μl) were electroporated into 20 μl MegaX DH10B T1 electrocompetent cells (Thermo Fisher Scientific) using a GenePulser II (BioRad; Settings: 2.0 kV, 200 Ω and 25 μF) in four replicates. Cells were resuspended in 2 ml S.O.C. recovery medium (Thermo Fisher Scientific) and incubated for 1 h at 37 °C and 250 rpm. Replicates were pooled and plated on two 245 mm × 245 mm plates (Corning) with ampicillin selection (100 μg/ml), which yielded 200 × (GeCKO sublibrary A) 800× (GeCKO sublibrary B) and 500× (Brunello) library coverage. After 14 h of incubation at 37 °C, colonies were scraped off and combined, and plasmid DNA was extracted using Nucleobond Xtra Maxi EF (Macherey Nagel).

**Lentivirus production**. To produce lentivirus, HEK 293 T/17 cells were seeded at 70% confluence in 48 ml Opti-MEM, GlutaMAX (Thermo Fisher Scientific) with 5% FBS and 0.2 mM sodium pyruvate in four T225 flasks/library 16 h before transfection. Transfections were performed using Lipofectamine 3000 and P3000 Enhancer (Thermo Fisher Scientific) according to manufacturer's instructions. In

short, for each transfection, 44 μg lentiCRISPRv2 plasmid library were mixed with packaging plasmids pMD2.G (22 μg, Addgene #12259) and psPAX2 (33 μg, Addgene #12260), 141 μl P3000 Enhancer and 6 ml Opti-MEM (Thermo Fisher Scientific). A mixture of 6 ml Opti-MEM and 165 μl Lipofectamine 3000 was added to the plasmid mixture before incubation for 20 min at room temperature. 24 ml medium was removed from the cells, before drop-wise addition of the lipid-DNA complex and incubation at 37 °C, 5% CO$_2$. After 6 h, the medium was replaced with 48 ml pre-warmed Opti-MEM, GlutaMAX with 5% FBS, 0.2 mM sodium pyruvate and 1% BSA (Roche). Cells were incubated at 37 °C, 5% CO$_2$. Virus supernatant was harvested at 24, 48 and 72 h post transfection, centrifuged at $2100 \times g$ at 4 °C for 10 min, and filtered through a 0.45 μm low protein binding membrane (Merck) and stored at 4 °C until all harvests had been completed. Finally, the virus was concentrated using PEG virus precipitation kit (BioVision; GeCKO sublibraries A and B) or ultracentrifugation at 84,000×g for 90 min at 4 °C (Beckman Coulter SW28; Brunello), resuspended in R10, aliquoted and stored at −80 °C.

**Virus titration and transduction**. Jurkat cells were transduced with GeCKO sublibraries A and B while H9, THP-1 and primary CD4$^+$ T cells were transduced with the Brunello library. Primary CD4$^+$ T cells were activated with CD3/CD28 beads one day prior to transduction as described above. To ensure a multiplicity-of-infection (MOI) around 0.3 to 0.4, to minimize the risk of multiple sgRNA integration in single cells, all cell types and virus batches were titrated to find the optimal virus volumes. $3 \times 10^6$ cells/well were seeded in 12-well plates in 2 ml medium supplemented with Polybrene (8 μg/ml; Millipore; Jurkat, THP-1) or Lentiboost (20 μl each of solution A and B; Sirion Biotech; H9, primary CD4$^+$ T cells). Different volumes of virus were added to each well except the no-transduction control well. The plate was centrifuged at 1,000×g for 1 h at 32 °C and incubated at 37 °C, 5% CO$_2$ overnight. After 24 h, cell lines were pelleted and resuspended in fresh medium, primary cells were left untouched. After 48 h, all cells were pelleted and resuspended at $5 \times 10^5$ cells/ml in fresh medium. 1 ml/well were seeded in two identical 12-well plates, and puromycin (1 μg/ml, Sigma) were added to one of the plates. The plates were incubated at 37 °C, 5% CO$_2$ for 72 h, then cells in all wells were counted using Trypan Blue (Thermo Fisher Scientific) exclusion. MOI were calculated for all virus volumes as (viable cell count in replicate with puromycin subtracted by viable cell count in no-transduction control) divided by viable cell count in replicate without puromycin.

Large-scale transductions were performed as above using virus volumes corresponding to MOI 0.3-0.4. Jurkat cells were transduced in three 12-well plates/sublibrary while H9 and THP-1 cells were transduced in four 12-well plates. Primary CD4$^+$ T cells from five donors were transduced in 2–3 plates depending on cell number. Wells were pooled into larger flasks when the transduction medium was replaced one (cell lines) or two (primary cells) days after spinfection. Puromycin selection started 48 h after spinfection and was maintained for 7 days. Cell lines were expanded and frozen before use for cell sorting. Sufficient cell number for 500× library coverage was maintained at all times. Primary cells were re-activated with Dynabeads Human T-Activator CD3/CD28 three days prior to sort.

**Flow cytometry**. Cell binding of test antibodies was analyzed by flow cytometry. For T cell analysis (Jurkat, H9 and primary CD4$^+$ T cells), human IgG1 antibodies were diluted to 10 μg/ml in PBS + 0.5% BSA and added to test cells, 50,000 cells/well in 25 μl reaction volume, and left to bind for 1 h at +4 °C. After washing, bound IgG was detected using an APC conjugated anti-human-IgG antibody (Jackson Immunoresearch # 109-136-098, 1:200 dilution) together with a live/dead cell marker (SYTOX green, Thermo Fisher Scientific) and analyzed in BD LSRFortessa (BD Biosciences) using BD FACSDiva Software v.8.0.2. Analysis of THP-1 cell binding was done essentially the same way, but cells were blocked with 10 mg/ml human IgG (Kiovig, Baxalta) for 15 min before addition of biotinylated test antibody, and bound IgG was detected with Streptavidin-APC (Miltenyi Biotec, 1:200 dilution). Murine control antibodies with known targets were conjugated and no secondary antibody was used (CD2-PE-Cy7 (BD Biosciences, #335821, 1:10 dilution) and CD45-APC (BD Biosciences, #555485, 1:5 dilution). Flow data was analyzed using FlowJo v.10.4.2.

**Fluorescence activated cell sorting (FACS)**. For sorting, 1 to $2 \times 10^7$ cells in 200 μl reaction volume were stained as above. Stained cells were washed and resuspended in 1–2 ml R10 medium. Live (SYTOX green negative) cells were sorted on BD FACSAria Fusion (BD Biosciences) with BD FACSDiva Software v.8.0.2 in a two-way sort with the 1.5–2% most APC negative cells (test cells) in one gate and the 20% most APC positive cells (control cells) in the other gate (for CD2 control antibody, the PE-Cy7 channel was used instead of APC). For full gating path, see Supplementary Fig. 4. Data was analyzed using FlowJo v.10.4.2. For primary CD4$^+$ T cells and PMA-polarized THP-1 cells, genomic DNA was extracted from sorted cells without pre-enrichment. Jurkat, H9 and non-polarized THP-1 cells sorted in the fluorescence-negative gate (pre-enriched cells) were cultured and expanded in medium supplemented with 100 U/ml penicillin and 100 μg/ml streptomycin (Thermo Fisher Scientific) for a second sort and expansion before DNA extraction.

For pre-enriched cells, cells from both sort rounds were analyzed by flow cytometry as above to follow the enrichment of negative cells between sorts.

We did not observe heterogeneous expression/staining in any of our cell line experiments, either upon thawing or after culturing. However, for primary in vitro-activated CD4+ T cells, we observed variation in the average antigen expression level between donors for some antibodies, as well as signs of phenotypic heterogeneity within some donors (Supplementary Fig. 5). Because of the variability in expression between and within donors, we used higher numbers of replicates when working with primary cells. Thus, transduced cell lines were sorted in three replicates. For in vitro-activated CD4+ T cells, we used five replicates, each of which represented independently transduced cells from a unique donor.

**Library preparation and massively parallel sequencing**. Genomic DNA (gDNA) was extracted using the QIAamp DNA Micro Kit (cells sorted once; 50,000–$1 \times 10^6$ cells), the QIAamp DNA Blood Mini Kit (pre-enriched, sorted and expanded cells; $5 \times 10^6$ cells) or Blood and Cell Culture DNA Midi kit (unsorted transduced cell lines; $4 \times 10^7$ cells), all from Qiagen. One-step PCR was performed to amplify lentiCRISPR sgRNAs from gDNA and attach Illumina adapters and indexes to the samples. Reaction volume was 50 μl/sample, with 25 μl NEBNext® Ultra™ II Q5® Master Mix (New England Biolabs) and 0.5 μM of each primer. Forward primers include a variable length sequence to increase library complexity, while reverse primers include an 8-bp index sequence to facilitate multiplexing (Supplementary Table 3). For unsorted cells, 200 μg gDNA (5 μg/reaction in 40 reactions) was used to achieve a 500-fold representation of each sgRNA. For sorted samples, all gDNA (50 ng–5 μg) was used. PCR amplification was carried out with 26 (<0.2 μg gDNA), 24 (0.2–1 μg gDNA) or 22 (>1 μg gDNA) cycles with the following conditions: 98 °C/30 s; 22–26 cycles of 98 °C/10 s, 65 °C/30 s, 72 °C/30 s; followed by 72 °C/5 min. PCR products were purified (DNA clean and concentrator-5, Zymo Research), quantified (Qubit™ dsDNA HS Assay Kit, Thermofischer Scientific) and analyzed for purity and size on Agilent 2100 Bioanalyzer using the Agilent 1000 DNA kit and Agilent 2100 Expert software version B.02.08.SI648(SRI). The samples were combined on eight flow cells (Illumina NextSeq 500/550 High Output Kit v2.5 (150 cycles), #20024907) with approximately 10× molar excess of unsorted samples over sorted samples and 25% PhiX added and sequenced on Illumina NextSeq 500 using single reads to a median depth of 4 million usable reads/sample. Basecalling and demultiplexing were done using bcl2fastq version 2.20.0.422, quality of resulting fastq files were then inspected using fastQC version 0.11.8. with default options. Genes whose sgRNA were enriched in the antigen-negative fraction relative to control cells were identified with a robust ranking aggregation algorithm implemented in the MAGeCK method[25]. sgRNA in individual samples were counted using the count command, and samples were tested against matched controls using the MAGeCK test command. Replicates were analyzed individually to test the correlation between replicates. The final gene enrichment score was calculated based on pooled analysis of all replicates.

**Calculation of MHC class I enrichment score**. To facilitate deconvolution of MHC class I antibodies, we defined a MHC class I enrichment score as the –log$_{10}$ P-value for one-sided Wilcoxon rank-sum test for the MAGeCK scores of 11 genes that have previously been shown to be required for, or modulate, the expression of MHC class I (*HLA-A*, *HLA-B*, *HLA-C*, *B2M*, *TAP1*, *TAP2*, *TAPBP*, *NLRC5*, *RFXAP*, *RFX5* and *RFXANK*) versus the MAGeCK scores for all other genes in the genome.

**Validation of inferred antibody specificity**. Antibody specificity was validated experimentally using either enzyme-linked immunosorbent assay (ELISA), flow cytometry with blocking using polyclonal antibodies, or ectopic overexpression of the target gene followed by flow cytometry.

For ELISA, target proteins (Supplementary Table 4) were coated to plates overnight at +4 °C. The next day antibodies were serially diluted 1:2 or 1:3 starting at 133 nM and left to bind the washed ELISA plate for 1 h at room temperature. Binding was detected using a horseradish peroxidase (HRP) conjugated anti-human-Fc antibody (Jackson ImmunoResearch #109-036-006, dilution 1:5,000), followed by a luminescent substrate (SuperSignal ELISA Pico, Thermo Fisher) and plates were read in a plate reader (Tecan Ultra with Tecan Magellan v.3.0).

For blocking experiments, cells were stained and analyzed with or without pre-block with a polyclonal antibody against the suggested antibody target (Supplementary Table 4). After incubation for 1 h at 4 °C with the block antibody (20 μg/ml), the test mAb was added (1 μg/ml) and incubation was continued for an additional 30 min before wash and addition of secondary antibody and flow cytometry analyzes as described above.

For overexpression experiments, cDNA for the main isoforms of *TMEM2*, *CD97*, *ITGAM* or *ITGB2* (GeneArt) were cloned into an expression vector and used for transfection of CHO-S cells using FreeStyleMAX Reagent (Thermo Fisher) according to manufacturer's instructions. *ITGAM* and *ITGB2* were co-transfected in order to express the Mac-1 heterodimer. *TMEM2-* and Mac-1-transfected cells were analyzed by flow cytometry 48 h after transfection, while 1.5 mg/ml of Geneticin (Thermo Fisher) was added to CD97 transfected cells as selection pressure. Surviving cells with high expression of CD97 were sorted by FACS, expanded, frozen and used in flow cytometry as described above.

**Essentiality analysis**. To understand if cell surface proteins are more, or less, essential than genes in general, we analyzed data from genome-wide lentiviral CRISPR-Cas9/sgRNA proliferation screens encompassing 436 cell lines from the Dependency Map (DepMap; Avana 18Q2; downloaded from the Broad Institute Cancer Dependency Map portal https://www.depmap.org/portal) on June 8 2018. To define the set of membrane protein genes, we used predicted membrane proteins (protein class: predicted membrane proteins) as defined Human Protein Atlas (www.proteinatlas.org) on July 6 2020. To define CD antigen genes, we used the list provided by Human Genome Nomenclature Consortium (gene group: CD molecules; www.genenames.org) on June 29 2020. Ribosomal protein-coding genes were defined as the genes encoding the known proteins of the small and large ribosomal subunits[55]. For enrichment testing, we used RenderCat software with default settings[50].

**Reporting summary**. Further information on research design is available in the Nature Research Reporting Summary linked to this article.

## Data availability

All relevant data are available from the authors. Source data are provided with this paper.

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

## Acknowledgements

The work was sponsored by research grants from the Swedish Foundation for Strategic Research (grant no. ID15-0012 to B.N.), Inga-Britt and Arne Lundbergs Research Foundation (grant no. 2017-0055 to B.N.), and Lund University.

## Author contributions

J.M., B.F., I.T. and B.N. conceived the project. J.M., F.J., E.E., R.A. and M.P. carried out experiments. L.E., J.M., A.N. and B.N. analyzed data. J.M. and B.N. wrote the manuscript. All authors contributed to the final manuscript.

## Funding

## Competing interests

J.M., I.T. and B.F. are employed by BioInvent International AB. The remaining authors declare no competing interest.
