## [Peer Review File · Nature Communications]

Reviewers' Comments:

Reviewer #1:

Remarks to the Author:

Mattsson et al. describe a genome editing workflow for antibody target identification. They deploy existing sgRNA libraries and CRISPR/Cas9 constructs to perform antigen ID for a set of ~30 antibodies. The identities of the hits from the screening were further validated by ELISA, blocking experiments and cellular over expression. The authors tackle a very important and difficult problem that has plagued the antibody research community for a long time, and the described toolkit/workflow represent a significant advance for the field. I have the following suggestions to improve the manuscript.

1. the authors should discuss the applicability of the technique to low abundance targets since signal to knockout will be much less discriminated and sorting will likely require several rounds of "enrichment" of non-expressors to see the gene set patterns arise in the sequencing data.
2. the authors need to add to the main text the number of lentivirally transduced cells that need to be sorted to maintain library coverage, oversample the library and not miss the target. These are key details to assess the limits of the technique.
3. Along these lines, the authors state in the discussion it should not be too hard finding a host that can be cultured and transduced. This is probably the case for cancer cell lines, but many examples of phenotypic screens take place on primary cells or tissues. Cloning these antigens can be difficult, so they at least need to acknowledge these situations in the discussion.
4. Figure 1e states "both replicates". This needs to be clarified, was this replicate sorts? library transductions? or sequencing? These would all have different meanings. Along these lines, were replicate sorts done and did they identify the identical/similar gene clusters?
5. Supplementary Figure 3 needs to indicate the units for antibody concentration on the x-axis.
6. While the library over expression strategies can be laborious as mentioned by the authors in the intro and discussion, they are no more laborious than the process described in this communication. In fact, the described workflow of multiplex expression cloning is quite similar and can be used for the identification of many antibody targets as well (e.g. Agarwal and Shusta, *Proteomics* 2009 and Agarwal Lippman and Shusta, *J Neurochem*, 2010). This is not to say that the authors methodology doesn't have its exciting advantages. The authors also point that the knockout methods described here identify other pathway machinery and this would be unlike expression cloning methods. They could also note that the additional knowledge about pathway machinery could itself identify new therapeutic targets (in addition to the cell surface receptors themselves). They should note these two items in the discussion.

Reviewer #2:

Remarks to the Author:

Summary

The authors use existing antibodies that are known to bind targets on cell lines but whose target is actually unknown yet (here, Jurkat or H9 cells), and use a genome-wide CRISPR-CAS library to randomly target genes. Starting by cells that are stained with the antibody of interest, by selecting modified cells that do not bind the antibody anymore, the authors aim to identify antibody targets. The technique seems suited for tumor antigens, when you know an antigen binds a tumor but the question is which one.

A powerful result of this method is to assess cross-reactivity of antibodies towards many antigens due to higher cross-reactivity. Here, cross-reactivity with many antigens was due to high MHC

affinity.

The bioinformatic method used for gene enrichment, MAGeCK seems to already exist, is not described, and the paper seems to be only generating the CRISPR library and applying the MAGeCK method, which is only an incremental benefit. Therefore, the only new thing is the proof of concept that genome-wide targeting can help finding the main antibody target (which is a great achievement in my opinion).

Personal opinion (not a revision requirement): The incremental progress of this study would benefit from extended validations to show its applicability in the context of tumor antigens; isolation of cross-reactive antibodies or experimental set up with primary cells.

Major

Jurkat cells are not stable for the expression of surface molecules, some heterogeneity in expression happens, typically when thawing new batches. Could the authors show on some antigen cases that the cells homogeneously express the marker? For instance, starting from cells binding with the antibody (the original batch), do the cells still all bind with the antibody after a few passes. How does this impact the result, in particular for the bioinformatic analysis?

[note: in supp Fig 3 the authors show staining with the candidate antibodies, and it seems \approx homogeneous for most antigens but not all. Can the authors discuss this].

Minor

Supp Fig 2: Why are candidates treated according to different thresholds (sometimes below 2, sometimes above 2), and how are the candidates treated when there are 2 candidates above 2? Are they always from the same pathway / expression machinery ? or is this really cross-reactivity.

Supp Fig 3 should be brought to the main text in a different format, this is the experimental validation-

Note: testing reactivity on an ELISA while the article shows the impact of MHC affinity on cross-reactivity might be of limited use when asking for developability of therapeutic antibodies not to be self-reactive. Any way to validate on at least one antigen using tetramers with multiple of the antibodies (supposed or not to bind this antigen)?

The authors show the recovery of antibody targets. Can the authors discuss to what extent they can differentiate between slightly different Abs that target different epitopes on the same target molecule?

To what extent is your approach sensitive to antibody cross-reactivity? Can you discuss how the results would look like for broadly reactive antibodies?

Response to comments

We thank the referees for their constructive comments. All points raised have been addressed. The changes are listed below, and also indicated in the revised manuscript.

Response to comments from Reviewer #1

“Mattsson et al. describe a genome editing workflow for antibody target identification. The authors tackle a very important and difficult problem that has plagued the antibody research community for a long time, and the described toolkit/workflow represents a significant advance for the field.”

THANKS. We thank Reviewer #1 for his/her encouragement. We are happy to hear that our work was well received.

“The authors should discuss the applicability of the technique to low-abundance targets since signal to knockout will be much less discriminated and sorting will likely require several rounds of enrichment of non-expressors.”

DONE. This is now addressed in Discussion (p.10-11). The approach investigated here does not seem to require high target abundance/staining, but seems to work also for targets of intermediate abundance/staining (*c.f.*, mAb31 and mAb37 in **Supplementary Fig. 1**). The lowest abundance/staining for which the approach works remains to be determined. However, one would expect that the lower the abundance is, the lower the signal-to-noise ratio will become, and that this can be partly overcome by increasing the number sorting/expansion rounds (**Fig. 1b-d**), as well as the number of replicates (*i.e.*, the statistical power).

“The authors need to add to the main text the number of lentivirally transduced cells that need to be sorted to maintain library coverage, oversample the library, and not miss the target. These are key details to assess the limits of the technique.”

DONE. This is now clarified in Results (p.5). To maintain library coverage, we used at least 100 cells per sgRNA in each sort (*i.e.*, at least 8 million Brunello-transduced or 13 million GeCKO-transduced cells).

“The authors state in the discussion it should not be too hard finding a host that can be cultured and transduced. This is probably the case for cancer cell lines, but many examples of phenotypic screens take place on primary cells or tissues. Cloning these antigens can be difficult, so they at least need to acknowledge these situations in the discussion.”

DONE. This has now been clarified in Results (p. 4-5) and Discussion (p. 11-12). We agree that many phenotypic screens take place on primary cells. For example, 29 of the 38 antibodies in our test set were identified through phenotypic screens on primary cells (the 23 Treg and the 6 TAM antibodies). Yet, this does not necessarily mean that the target deconvolution needs to be done on primary cells. As exemplified in our study, it is often possible to find a cell line that is antigen-positive and can be used for target deconvolution. We also exemplify that our approach can in principle be applied to primary cells (in our case *in vitro*-activated CD4⁺ T cells from healthy blood donors; mAb18 to mAb23). If no antigen-positive, practically useable test cell can be identified, our approach is not applicable.

“Figure 1e states ‘both replicates’. This needs to be clarified, was this replicate sorts? library transductions? or sequencing? These would all have different meanings. Along these lines, were replicate sorts done and did they identify the identical/similar gene clusters?”

DONE. The nature of the replicates has been clarified in Results (p.6) and the legend of **Figure 1**. We used sort replicates from the same transduced pool for cell lines, and transduction replicates with different donors for primary cells. For most antibodies, we observed high positive correlation in gene enrichment scores between replicates.

“Supplementary Fig. 3 needs to indicate the units for antibody concentration on the x-axis.”

DONE. Units for antibody concentration (nM) have been added to **Supplementary Fig. 3**.

“While the library overexpression strategies can be laborious as mentioned by the authors in the intro and discussion, they are no more laborious than the process described in this communication. In fact, the described workflow of multiplex expression cloning is quite similar and can be used for the identification of many antibody targets as well (e.g. Agarwal and Shusta, Proteomics 2009 and Agarwal Lippman and Shusta, J Neurochem, 2010). This is not to say that the authors’ methodology doesn’t have its exciting advantages. The authors also point that the knockout methods described here identify other pathway machinery and this would be unlike expression cloning methods. They could also note that the additional knowledge about pathway machinery could itself identify new therapeutic targets (in addition to the cell surface receptors themselves). They should note these two items in the discussion.”

DONE. This has now been clarified in Discussion (p.11). Our approach offers advantages over library overexpression techniques. Firstly, these techniques lack the ability to identify antibodies towards protein complexes, or obtain insight into the underlying pathways, through dependencies the way pooled CRISPR/Cas9 does. Secondly, arrayed overexpression reagents are hard to make, practically only available at a high cost via commercial services, and it is well known that the chance of success is considerably lower than in our study. Thirdly, pooled cDNA library reagents also pose challenges that are not present with Cas9/sgRNA libraries, including being more difficult to synthesize, transfect and deconvolute. Finally, while pooled CRISPR-dCas9/sgRNA overexpression libraries have been developed, these produce effect sizes that are significantly weaker than complete knockouts.

Response to comments from Reviewer #2

“Proof of concept that genome-wide targeting can help finding the main antibody target, which is a great achievement in my opinion.”

THANKS. We thank Reviewer #2 for his/her encouragement. We are happy to hear that our work was well received.

“Jurkat cells are not stable for the expression of surface molecules, some heterogeneity in expression happens, typically when thawing new batches. Could the authors show on some antigen cases that the cells homogeneously express the marker? For instance, starting from cells binding with the antibody (original batch), do the cells still all bind with the antibody after a few passes. How does this impact the result, particularly the bioinformatic analysis?”

DONE. This has now been clarified (Methods; p.17-18). We did not observe heterogeneous expression/staining in any of our cell line experiments, either upon thawing or after culturing. However, for primary in vitro-activated CD4⁺ T cells, we observed variation in the average antigen expression level between donors for some antibodies, as well as signs of phenotypic heterogeneity within some donors (exemplified in **Supplementary Fig. 4**). Because of the variability in expression between donors, we used higher numbers of replicates when working with primary cells. Thus, transduced cell lines were sorted in three replicates. For in vitro-activated CD4⁺ T cells, we used five replicates, each of which represented independently transduced cells from a unique donor.

“Supp Fig 2: Why are candidates treated according to different thresholds, and how are the candidates treated when there are 2 candidates? Are they always from the same pathway / expression machinery?”

DONE. The criteria used to prioritize candidate genes are detailed on p.6-7. Briefly, we use three criteria. Firstly, we prioritize all surface protein genes with FDR < 5%. Secondly, we identify MHC class I-targeting antibodies based on enrichment of genes required for MHC class I expression. Thirdly, for antibodies where no target was identified using the first two criteria, we prioritize the surface protein gene with the lowest FDR, even if it is higher than 5%. The criteria used for each of the test antibodies are listed in **Supplementary Table 1**. We found several of genes within the same expression machinery, mainly MHC class I and also integrins (**Fig. 2** and **Supplementary Fig. 2**).

“Can the authors discuss to what extent they can differentiate between slightly different Abs that target different epitopes on the same target molecule?”

DONE. This has now been commented in Discussion (p.12). Because our approach is based on gene knock-out using CRISPR/Cas9, it cannot differentiate between slightly different antibodies that target different epitopes on the same target molecule.

“To what extent is your approach sensitive to antibody cross-reactivity? Can you discuss how the results would look like for broadly reactive antibodies?”

DONE. This is now commented in the Discussion (p.12). Broadly reactive antibodies will be difficult to deconvolute as the target genes are knocked-out one at the time, and the test cells will therefore never turn completely antigen-negative.

“Supp Fig 3 should be brought to the main text.”

We understand the referee’s request. Yet, we prefer to keep **Supplementary Fig. 3** in the supplement, as moving it to the main text would require us to break up Figure 2 into multiple different figures, which we believe would reduce the quality of the presentation.

Additional changes

To comply with format guidelines, we have made the following changes:

1. Subheadings added.
2. Order of sections corrected.
3. Data availability statement added.

Reviewers' Comments:

Reviewer #1:

Remarks to the Author:

The revisions are appropriate and address my concerns.

Reviewer #2:

Remarks to the Author:

The authors have addressed all comments.